# A Rare Presentation of Metastatic Renal Cell Carcinoma Masquerading as Vitritis: A Case Report and Review of Literature

**DOI:** 10.3390/diagnostics12071712

**Published:** 2022-07-14

**Authors:** Maria Del Valle Estopinal, Robert T. Swan, Kevin Rosenberg

**Affiliations:** 1Ophthalmic Pathology Division, Departments of Ophthalmology and Pathology, University of California, Irvine, CA 92868, USA; 2Department of Ophthalmology & Visual Sciences, SUNY Upstate Medical University, 750 East Adams Street, Syracuse, NY 13210, USA; rswanmd@gmail.com (R.T.S.); krosenbe22@gmail.com (K.R.); 3Retina-Vitreous Surgeons of Central New York, PC, 200 Greenfield Parkway, Liverpool, NY 13088, USA

**Keywords:** metastatic renal cell carcinoma, vitritis, tyrosine kinase inhibitor, retinal vasculitis

## Abstract

We report a case of a 74-year-old gentleman who presented with floaters and decreased vision in the right eye after cataract surgery. His past medical history was significant for metastatic renal cell carcinoma (mRCC) to bone, lung and abdomen which was presumed stable for the last two years while on the tyrosine kinase inhibitor (TKI), pazopanib. Clinical examination revealed significant vitritis with a distinctive clumping of cells on the pre-retinal surface and posterior hyaloid face. Magnetic resonance imaging of the brain revealed new lesions suspicious for metastases. A diagnostic vitrectomy was performed to determine the nature of the vitritis and clear the visual axis. Cytopathologic evaluation of the vitreous demonstrated clusters of malignant cells that were positive for AE1/AE3 and PAX-8, and negative for the CD20, CD3, RCC, SOX-10 and S-100 immunohistochemical markers. The overall findings favored a metastatic RCC to the vitreous. Choroidal and retinal metastases from mRCC have been previously reported; however, vitreous involvement by mRCC with no evidence of retinal or choroidal mass has not been described. New treatments of mRCC include TKIs which target vascular endothelial growth factor receptors (VEGFRs). Herein, we analyze the factors that could have precipitated this unusual metastasis to the vitreous.

## 1. Introduction

Metastases to the ocular structures are rare and occur by hematogenous spread. Metastatic disease can be present in any part of the eye or orbit, but the uveal tract is the most common structure involved due to its robust vascular supply. The perimacular choroid is the principal targeted area due to its increased vascular supply [1]. Based on a literature review, the metastases to the eye are usually detected in the choroid (88%), iris (9%), or ciliary body (2%), and, rarely, in the retina (<1%) [2]. The most common primary tumors to metastasize to the eye are breast (47%), lung (21%), and the gastrointestinal tract (4%) [3,4].

Isolated metastases to the retina and vitreous are extremely rare. Shields et al. [2] have previously reported eight cases of retinal metastasis, of which four originated from cutaneous melanoma. In the same study, the retinal metastases were unilateral and unifocal, showing a tumor in seven of the cases. Clinically, metastases to retina/vitreous from carcinoma appear “white with perivascular infiltration,” mimicking an infectious or inflammatory retinitis or forming a retinal mass [2,5]. The tumor cells may initially metastasize to the inner retina, producing secondary vitreous seeding. Other possible pathways of vitreous invasion are from a choroidal metastasis that has invaded the retina or from a tumor in the central nervous system that spreads through the optic nerve [6].

RCC is a rare malignancy accounting for 2–3% of systemic tumors [7]. It occurs more often in men during the seventh and eighth decade of life. Thirty percent of patients with RCC have metastatic disease at the initial diagnosis [8]. The most common sites of involvement are lung, bones, liver, and brain. Metastatic RCC to the eye, orbit and periocular region are unusual, with less than 80 cases reported in the English literature [8]. In a large series of 227 cases of metastatic disease to the eye, Ferry and Font [9] reported that 3% of metastases were from RCC. New treatment modalities for the management of advanced RCC have been approved including antiangiogenic tyrosine kinase inhibitors (TKIs) and immune checkpoint inhibitors (ICIs), among others with successful survival benefits.

Herein, we present a case of mRCC to the eye masquerading as a severe vitritis and discovered one month after cataract surgery in a patient with prior history of a TKI therapy.

## 2. Case Presentation

A 74-year-old gentleman was referred to the Department of Ophthalmology at Upstate Medical University for vitritis and decreased vision in the right eye one month following uneventful cataract surgery in that eye.

His past medical history was significant for cutaneous melanoma in-situ of upper extremity (resected 54 months prior to current visit), a pT3aN0 clear cell renal cell carcinoma of the right kidney (s/p right radical nephrectomy), and a pT2aNx papillary renal cell carcinoma type I of the left kidney (s/p partial nephrectomy). The latter carcinomas were diagnosed four years before current presentation and followed by multiple metastases to bone, lung, and abdomen. Following the detection of the metastases, the patient was immediately started on pazopanib.

On ophthalmologic examination, visual acuity of the right eye was 20/800 and 20/30 in the left eye. Inflammation was graded according to the 2005 Standardization of Uveitis Nomenclature scale [10]. The right eye was notable for 1 + cellular reaction of the anterior chamber, a well-positioned posterior chamber intraocular lens, absence of cells in the anterior vitreous but posterior 3 + vitritis composed of large opacities and a deep cyst-like structure. B-scan ultrasound of the right eye demonstrated increased reflectivity of the posterior hyaloid face and suggested the “cyst” was in fact a Weiss ring (Figure 1A). No intraocular mass or retinal detachment was observed. The intravenous fluorescein angiogram (IVFA) of the right eye was limited by blockage from the vitritis but showed diffuse patchy hyperfluorescence during late phase with areas of hyperfluorescence along the venous arcades suggestive of mild retinal vasculitis (Figure 1B). The left eye exam was notable for a mild nuclear cataract and posterior vitreous detachment but no evidence of inflammation or other abnormalities on diagnostic imaging. A review of symptoms was significant for new-onset cognitive impairment and balance issues. At that moment, the differential diagnosis of the vitritis in the right eye included primary vitreoretinal lymphoma, metastatic disease, and infectious process. However, infectious endophthalmitis was felt to be unlikely due to a lack of pain or ocular injection on exam.

The patient was referred to his oncologist for screening of the central nervous system disease potentially secondary to lymphoma or renal cell carcinoma. Magnetic resonance imaging of the brain revealed heterogeneous enhancing lesions in the cerebellum, brainstem, and right hemisphere with associated hemorrhage. A neuro-oncology consultation reported that the brain lesions were most consistent with metastatic renal cell carcinoma. Radiation oncology recommended radiotherapy targeting the brain lesions, discontinuing pazopanib, and initiating the programmed death receptor-1 (PD-1) inhibitor, nivolumab.

Due to the diagnostic uncertainty and the significant visual impairment caused by the vitreous opacities, the patient underwent diagnostic vitrectomy five days after presentation. The cytopathology department received one milliliter of clear colorless viscous fluid from undiluted vitreous specimen. Two cytocentrifuge slides were prepared for Diff-Quik staining preparation and one cytocentrifuge slide for Papanicolaou (Pap) stain. In addition, two 5 milliliter-syringes of clear, colorless fluid of dilute vitreous specimen were sent for flow cytometry analysis and microbiology studies, respectively.

Microscopically, sheets of malignant epithelioid cells forming three dimensional groups were identified in the vitreous. Some groups appeared to form micropapillary structures with enlarged, hyperchromatic, eccentrically located nuclei with small nucleoli and a moderate amount of cytoplasm (Figure 2). Immunohistochemical studies were performed with appropriate positive controls, encompassing anti-cytokeratin cocktail AE1/AE3 (mouse monoclonal antibody; 1:100–1:500 dilution range; Cell Marque, Rocklin, CA, USA); PAX8 (mouse monoclonal antibody; 1:50–1:200 dilution range; Cell Marque, Rocklin, CA, USA), CD20 (mouse monoclonal antibody; 1:250 dilution; ThermoFisher Scientific, Riverside, CA, USA), CD3 (mouse monoclonal antibody; 1:10–1:20; ThermoFisher Scientific, Riverside, CA, USA), RCC (Human clear renal cell carcinoma antibody, mouse monoclonal antibody; 150–200 microliter; ThermoFisher Scientific, Riverside, CA, USA), SOX10 (rabbit polyclonal antibody; 1:25–1:100 dilute; Cell Marque, Rocklin, CA, USA) and S100 (mouse monoclonal antibody; 1–2 mg/mL; ThermoFisher Scientific, Riverside, CA, USA). All antibodies and testing were performed in a CLIA-certified laboratory.

The tumor cells were positive for AE1/AE3 and PAX-8 stains and were negative for CD20, CD3, RCC, SOX-10 and S-100 protein (Figure 3). A flow cytometry analysis indicated that there was no evidence of leukemia or lymphoma. The cytomorphologic features observed in the vitreous sample shared similarities with those seen in prior histopathologic sections of the nephrectomy specimens. The combined clinical, cytopathologic and immunohistochemical findings were consistent with metastatic carcinoma of renal cell origin.

At the follow-up, one week after vitrectomy, the vision in the right eye had improved to 20/60. On clinical examination, the anterior chamber and vitreous each had 1+ cell, and an inferior vitreous base opacity was noted. OCT of the right macula showed a sawtooth like aggregation of cells on the pre-retinal surface with an intact foveal depression (Figure 1C). Choroidal thickness appeared normal and symmetric between the two eyes. The IVFA of the right eye was repeated to better evaluate the fundus and demonstrated diffuse patchy hyperfluorescence with hypofluorescence adjacent to the retinal arteries, optic nerve staining, and mild angiographic macular edema in the late phase (Figure 1D). There continued to be areas of venous hyperfluorescence seen best along the superior-temporal arcade suggestive of mild retinal vasculitis.

The patient developed vitreous hemorrhage in the right eye, one month postoperatively causing light perception vision and precluding an accurate evaluation of the fundus and detection of possible therapy-related changes. Three months after the vitrectomy, the patient presented clinical findings consistent with neovascular glaucoma. Bevacizumab intravitreal injection was recommended. The patient’s condition deteriorated, and he passed away six months later.

## 3. Discussion

RCC continues to represent an increasing proportion of the cancer landscape. This steady rise, continuing over the past 20 years, reflects an increase in the radiographic identification of incidental masses. However, RCC mortality has not declined in that time frame, implying that the increase in diagnoses is not solely due to early detection. RCC has risen to become the 7th most common cancer in men and the 8th most common cancer in women in the United States [11]. About 17% of patients with RCC have metastatic disease on initial presentation, with lungs being the most common metastatic site [11]. Uveal metastases from the kidney have been found rarely. Shields et al., [3] in their survey of 520 eyes with uveal metastases, report that 2% of the cases came from the kidney.

The biology of RCC is characterized by demonstration of VHL-associated molecular pathways. Patients with germline mutations in the von Hippel-Lindau (VHL) tumor suppressor are affected by a rare familial tumor syndrome, which is characterized by the predisposition to develop highly vascularized tumors in multiple organs. These include hemangioblastomas of the retina and central nervous system, renal cancer of the clear cell type (CC-RCC), and pheochromocytomas. A major function of pVHL is serving as the substrate recognition component of an E3 ubiquitin ligase, which ubiquitinates and targets the α-subunit of hypoxia-inducible factor (HIF) for oxygen-dependent proteolysis. Mutation or loss of VHL expression results in HIF-α stabilization, increased HIF transcriptional activity and the up-regulation of HIF target genes such as vascular endothelial growth factor (VEGF), glucose transporter 1 (GLUT-1) and erythropoietin (EPO), irrespective of oxygen levels [12]. Furthermore, a gene expression and genomic analysis of sporadic and VHL disease-associated renal masses have found a high degree of similarity between the VHL disease tumors and a homogenous subgroup of the sporadic tumors.

Clear cell renal cell carcinoma (ccRCC) is associated with chromosome 3p deletion in 74% of the cases and in 25% of the cases with mutation in the VHL gene. Papillary RCC (PRCC) is characterized by the presence of a papillary or tubulopapillary architecture, with neoplastic cells overlying a delicate fibrovascular core or forming compact tubules. PRCC occurs in sporadic or hereditary forms. Sporadic PRCC is the second most common carcinoma of the kidney, comprising 10–15% of renal neoplasms [13]. Two subtypes of PRCC have been described, I and II, which differ in genotype. Type I tumors show chromosome 7p and 17p gains, and type 2 tumors show allelic imbalance of one or more of chromosomes 1p, 3p, 5, 6, 8, 9p, 10, 11, 15, 18, and 22. In addition, bilateral synchronous renal cell carcinomas with different histology have been reported in the literature with approximately 6% of the cases presenting as ccRCC with contralateral PRCC, which was also observed in our case [14].

Immunohistochemically, most clear cell or PRCC (84%) are positive for RCC marker (glycoprotein in renal proximal tubular brush border) [11], PAX-8 (transcription factor critical for the development of eye, thyroid, urinary and reproductive organs), paired-box 2 (PAX-2) and cytokeratin AE1/AE3. Other markers can be used to differentiate each of the subtypes of renal cell tumors such as carbonic anhydrase IX (CAIX), vimentin and CD10 for clear cell renal cell carcinoma and CK7, Alpha-methyacyl-CoA racemase (AMACR) and CD117 for PRCC.

Herein, we describe a rare presentation of vitreous involvement by RCC in a background of metastatic disease to brain, lung and abdomen, and no definite evidence of retinal or choroidal tumor. The tumor cells were immunoreactive for pan-cytokeratin and PAX-8 stains. The latter has been shown to be useful for the diagnosis of mRCC [15,16]. The lack of expression of RCC immunohistochemical marker in the neoplastic cells could be related to changes in tumor antigenicity secondary to the vitreous microenvironment.

The IFVA prior to the diagnostic vitrectomy revealed features suggestive of retinal vasculitis which could have allowed vitreous seeding of RCC cells; however, the possibility of the spread of neoplastic cells through the optic nerve from foci of brain metastases is not entirely excluded. Systemic and local intraocular conditions are associated with retinal vasculitis including infectious processes, systemic inflammatory diseases, and malignancies, among others.

RCC is a well-known malignancy for inducing upregulated angiogenesis and hemorrhagic metastases related to alterations in pVHL [1,12]. Furthermore, the tendency of hemorrhagic retinal detachment secondary to choroidal metastasis as well as the propensity toward bleeding of RCC brain metastases have previously been described [17,18]. Published reports of intraocular hemorrhage in RCC are sparse. However, ocular oncologist VML Cohen reported it was her experience that renal metastases to the eye were more likely to cause intraocular hemorrhage [1].

Different treatment modalities for the management of RCC have been developed, including radical nephrectomy, laparoscopic partial nephrectomy, and new therapeutic regimens. Much attention is given to the use of VEGF-receptor tyrosine kinase inhibitor therapy and immune checkpoint inhibitors (ICI) as monotherapy as well as combined therapy for this malignancy. RCC is a highly angiogenic and immunogenic tumor, and new evidence suggests a potential beneficial synergistic effect of combined treatment modalities encompassing TKIs and ICIs (PD-1 and CTLA-4). The latter is based on improved progression-free survival or/and overall survival outcomes [19].

Our patient had received treatment with pazopanib prior to the development of vitritis. Pazopanib is a small-molecule tyrosine kinase inhibitor of growth factor receptors associated with angiogenesis and tumor cell proliferation. Its mechanism of action targets the inhibition of vascular endothelial growth factor receptors (VEGFR-1, 2 and 3), fibroblast growth factor receptors and transmembrane glycoprotein receptor tyrosine kinase (c-Fms), among others. Although effective disease control has been obtained as a first-line treatment; late side effects, disease control issues, and disease resistance have been described in these patients [19]. Li et al. [20] suggest on their paper that a patient with mRCC developed central retinal vein occlusion and fundus hemorrhage secondary to TKI therapy. Furthermore, pazopanib-induced cutaneous leukocytoclastic vasculitis has been reported as an adverse reaction to therapy [21].

The clinical presentation and outcomes of our case suggest that more personalized therapies should be considered for advanced RCC, targeting the different pathways involved in the development of this malignancy.

## 4. Conclusions

The diagnosis of mRCC is challenging, especially in small biopsies because of the variety of histologic appearance and clinical presentations. It is known to produce hemorrhagic metastases secondary to abnormalities in pVHL. We believe that factors such as vascular alterations associated with the pathophysiology of RCC, spreading of tumor cells from the brain through the optic nerve, and a possible failure of TKI therapy have contributed to the development of this unusual presentation of mRCC to the eye. Further studies are necessary to understand the pathogenesis of metastatic disease to the vitreous.

## Figures and Tables

**Figure 1 diagnostics-12-01712-f001:**
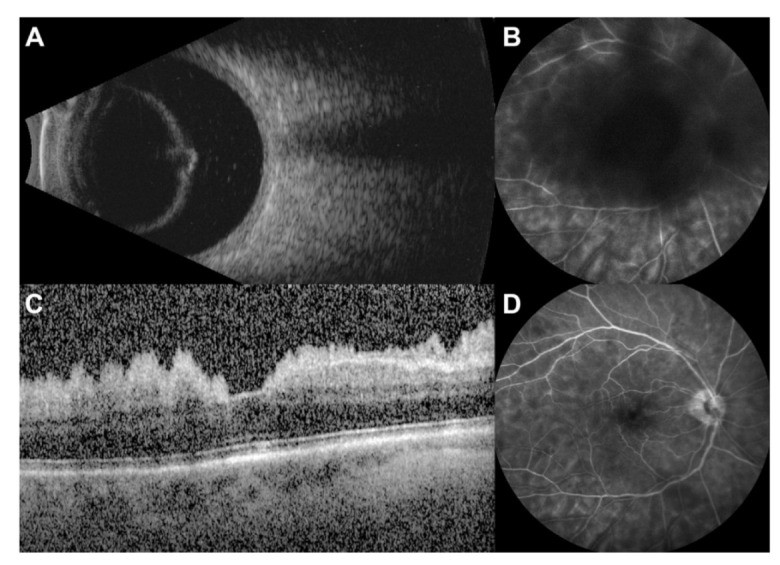
(**A**)**.** B-scan ultrasound of the right eye showing a posterior vitreous detachment with delineation of the posterior hyaloid face and vitritis caused by metastatic cells. (**B**)**.** Late-phase intravenous fluorescein angiogram of the right eye demonstrating blockage from the overlying vitritis. There is diffuse, patchy hyperfluorescence with areas of hyperfluorescence along the venous arcades suggestive of mild retinal vasculitis. (**C**)**.** Ocular coherence tomography (OCT) of the right macula obtained nine days after diagnostic vitrectomy demonstrating a sawtooth pattern of metastatic cells on the pre-retinal surface. (**D**)**.** Late-phase intravenous fluorescein angiogram of the right eye obtained nine days after diagnostic vitrectomy. There is diffuse, patchy hyperfluorescence, optic nerve staining, and mild angiographic macular edema. There is hypofluorescence adjacent to the retinal arteries and areas of venous hyperfluorescence seen best along the superior-temporal arcade that are suggestive of mild retinal vasculitis.

**Figure 2 diagnostics-12-01712-f002:**
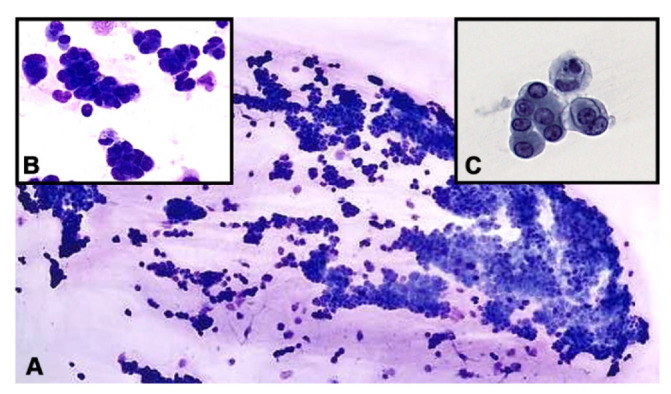
(**A**)**.** Hypercellular vitreous fluid demonstrating sheets of cohesive malignant epithelioid cells (Diff-Quik stain. Original magnification ×100). Inset (**B**)**.** Groups of tumor cells forming micropapilla-like structures (Diff-Quik preparation, ×400). Inset (**C**)**.** Pap-stained smear depicting clusters of neoplastic epithelioid cells with eccentric nuclei and vacuolated cytoplasm (original magnification ×400).

**Figure 3 diagnostics-12-01712-f003:**
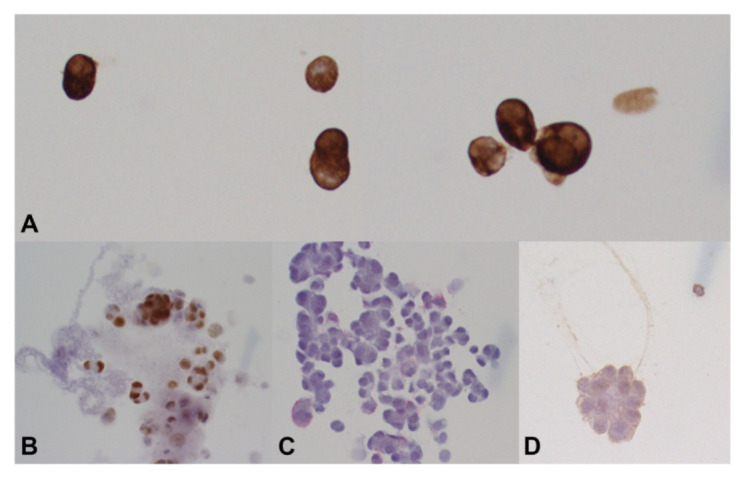
Immunohistochemical studies. Neoplastic cells depicting positive cytoplasmic staining for AE1/AE3 stain (**A**) and nuclear staining with PAX-8 (**B**). Negative expression for SOX10 (**C**) and S100 protein (**D**) was noted in tumor cells. Original magnification ×400 (**A**) and ×200 (**B**–**D**).

## Data Availability

Not applicable.

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
