# Peer review of "A Rare Presentation of Metastatic Renal Cell Carcinoma Masquerading as Vitritis: A Case Report and Review of Literature"

_diagnostics, 2022, doi:10.3390/diagnostics12071712_

Round 1

Reviewer 1 Report

This is a high quality case report with beautiful figures and histopathology images.  Since the patient had documented history of RCC and prior RCC resections, it would be interesting comment on similarities or dissimilarities between the current vitreous metastasis histopathology and prior RCC histopathology.  Did cells look same?  Same immunoprofile?  Any changes with the treatment?

Author Response

Thank you for your comments and considering our manuscript for publication at Diagnostics. The histopathologic sections of the prior nephrectomy specimens were reviewed after receiving the vitreous sample.  The cytomorphologic features of the cells in the vitreous shared similarities to those seen in the radical nephrectomy and partial nephrectomy histopathologic sections.  However, the vitreous is an specialized avascular tissue (composed of 99% water, collagen, glycoproteins, and soluble proteins) that behaves as a challenging environment for the tumor cells.  Therefore, we believe that the tumor cells morphology and structure might have changed to survive the conditions of the microenvironment.  Immunohistochemically, the tumor cells were positive for PAX-8 stain which has been recognized as useful diagnostic marker for metastatic RCC.  The vitreous/ocular structures microenvironment could have altered the tumor antigenicity explaining the lack of expression for RCC immunohistochemical study. Further studies are necessary to elucidate the mechanisms involved in ocular metastases as well as the changes induced by new cancer treatment modalities. 

Reviewer 2 Report

An interesting case of metastatic renal cell carcinoma is described in the article. The findings are well described and discussed in detail with relevant references.

In  165-166 lines, "RCC has risen to become the 7th most common cancer in men and the 8th most  common cancer in women in the US10", in this sentences, is "US10" correct? would it be "USA" correctly?

Author Response

Dear reviewer.

Thank you for your recommendation.  The recommendation has been applied in the manuscript.

Reviewer 3 Report

The authors of the entitled, “A Rare Presentation of Metastatic Renal Cell Carcinoma Masquerading as Vitritis: A Case Report and Review of Literaturepresent a case of cell carcinoma with adverse clinical results. It is an interesting case and in general well written although somewhat too long.  A few comments are listed below.

Lines 161-on. Since in the abstract the renal cell carcinoma is defined as RCC, there no need to do it again, except in the Introduction.

Lines 84-87, the left or right eye?  Please, clarify.

Lines 146-153: confusing here, are these findings, the same as those reported before under treatment?  Then, no need to relate them again.

 Also, were the signs and symptoms expected to improve?  That is most important for future and similar cases.

On the other hand, line 146, relates to “that post-surgery results” are the author referring to the “diagnostic vitrectomy”? Is that correct?  Please, clarify.

With such detailed results, the authors could do the discussion along the results or shorten the discussion by 2/3.

Author Response

Thank you for your recommendations.  The recommendations have been applied in the manuscript.

Regarding your question "were the signs and symptoms expected to improve?"

The prognosis of this patient with metastatic RCC to multiple organs including the eye was uncertain.  This is a rare presentation of metastatic RCC to the eye and unfortunately the hemorrhagic nature of the metastasis didn't allow us to monitor the intraocular structures during treatment. We believe that more personalized therapies for advanced RCC should be implemented based on the pathogenesis of this tumor and therefore evaluating the clinical outcome.